# Topological-Insulator-Based Gap-Surface Plasmon Metasurfaces

Andreas Aigner [1], Stefan A. Maier [1,2] and Haoran Ren [1,3,*] ✆

1. Chair in Hybrid Nanosystems, Nanoinstitute Munich, Faculty of Physics, Ludwig-Maximilians-Universität München, 80539 München, Germany; andreas.aigner@physik.uni-muenchen.de (A.A.); stefan.maier@physik.uni-muenchen.de (S.A.M.)
2. Department of Physics, Imperial College London, London SW7 2AZ, UK
3. MQ Photonics Research Centre, Department of Physics and Astronomy, Macquarie University, Macquarie Park, NSW 2109, Australia
* Correspondence: haoran.ren@mq.edu.au

**Abstract:** Topological insulators (TIs) have unique highly conducting symmetry-protected surface states while the bulk is insulating, making them attractive for various applications in condensed matter physics. Recently, topological insulator materials have been tentatively applied for both near- and far-field wavefront manipulation of electromagnetic waves, yielding superior plasmonic properties in the ultraviolet (UV)-to-visible wavelength range. However, previous reports have only demonstrated inefficient wavefront control based on binary metasurfaces that were digitalized on a TI thin film or non-directional surface plasmon polariton (SPP) excitation. Here, we numerically demonstrated the plasmonic capabilities of the TI $Bi_2Te_3$ as a material for gap–surface plasmon (GSP) metasurfaces. By employing the principle of the geometric phase, a far-field beam-steering metasurface was designed for the visible spectrum, yielding a cross-polarization efficiency of 34% at 500 nm while suppressing the co-polarization to 0.08%. Furthermore, a birefringent GSP metasurface design was studied and found to be capable of directionally exciting SPPs depending on the incident polarization. Our work forms the basis for accurately controlling the far- and near-field responses of TI-based GSP metasurfaces in the visible spectral range.

**Keywords:** topological insulator; gap–surface plasmon metasurface; $Bi_2Te_3$; MIM metasurface; beam steering; SPP excitation

## 1. Introduction

In the last two decades, tremendous progress has been made in understanding the fundamental physics of the combined fields of plasmonics and metamaterials and in utilizing them for a great variety of applications. Some prominent applications include beam steering [1], optical sensing [2], and displays [3,4]. However, the scope of plasmonic metamaterials is often limited by energy dissipation, especially in the UV-to-visible spectral range. Traditional noble metals, such as gold and silver, suffer from high losses due to interband transitions [5]. Furthermore, they are CMOS (complementary metal-oxide semi-conductor)-incompatible and show low thermal stability.

A new promising material class that is capable of overcoming these problems are 3D topological insulators (TIs). In recent years, these chalcogenide compounds, including $Bi_2Se_3$, $Bi_2Te_3$, and $Sb_2Te_3$, have attracted much attention in the condensed matter community for applications ranging from quantum computing to spintronics [6]. TIs exhibit strong spin–orbit coupling, leading to highly conducting symmetry-protected surface states. The band structure of these states is characterized by a single Dirac cone while the bulk has an ordinary insulating bandgap. In contrast to graphene, where the momentum and pseudo-spin of the carriers are locked, in TIs, the momentum and real spin are locked [7]. They were introduced to the field of plasmonics in 2013 by the observation of temperature-robust Dirac plasmons in TI gratings in the THz range [8]. By varying the grating width and film thickness, several groups [9,10] were able to map the dispersion relation of the

investigated plasmons, confirming their topological nature. Due to the subwavelength thickness of most studied TIs in the THz region, plasmons are excited at the top and bottom layers, simultaneously yielding a coupled system with acoustic (dark) and optical (bright) plasmon modes, which are purely spin- and charge-like, respectively [10–13]. The broad plasmon modes can interact with narrow phonon modes and result in the Fano-type line shapes [14,15]. Dirac plasmons show outstanding long lifetimes, high tunability in the THz range, temperature stability up to room temperature, and a large mode index compared to traditional metals [9]. Particularly interesting applications for TIs are THz photodetectors, which utilize plasma-wave oscillations in metal–TI–metal heterostructures [16] and field-effect transistors [17].

It is worth mentioning that not all charge carriers involved in a Dirac plasmon are massless Dirac electrons. In fact, it is a combination of Dirac electrons and massive electrons constituted from a non-topological 2D electron gas and bulk electrons [13,18,19]. This effect can be exploited by photo-doping TI gratings for ultra-high modulation depths above 2400%, as demonstrated by Sangwan Sim et al. [19]. Typically, Dirac plasmons are studied in the THz regime, where the bulk acts as a dielectric with a relatively low loss. When shifting to higher energies, which enable strong interband transitions, the bulk acts as if it were "optically metallic," meaning that the real part of the complex dielectric function is negative while the imaginary part is sufficiently small [20,21]. There have been multiple reports on plasmonic resonances with bulk TIs in the visible regime for applications ranging from nanometrology to holography [22–26]. Due to low losses at small wavelengths close to and in the UV, TIs exhibit superior plasmonic properties that outperform conventional noble metals, including gold and silver [23,27].

A widely used and well-established metasurface geometry is the gap–surface plasmon (GSP) metasurface. It consists of a dielectric spacer layer that separates an optically dense metallic film from an array of subwavelength metallic antennas forming a metal–insulator–metal (MIM) sandwich [28]. Combining a fairly easy fabrication with a high tunability of their optical response makes GSP metasurfaces well-suited candidates for flat lenses [29], holograms [30], sensing [31], perfect absorbers [28], and even surface plasmon polariton couplers [32]. However, conventional GSP metasurfaces dramatically lose their efficiency in the UV and blue wavelength range due to the high absorption loss of noble metals in this region. Even though switching to aluminum relieves the spectral restriction [33], it suffers from strong oxidization, which corrodes the material in the presence of air.

In this study, we numerically investigated TI GSP metasurfaces via finite element method simulations. Starting with a comparison of the plasmonic capabilities of the TI $Bi_2Te_3$ [23] and the conventional metals gold and silver [34], the spectral region of the superior plasmonic performance was identified. $Bi_2Te_3$ showed a higher plasmonic figure of merit (FOM) below 700 nm and 450 nm than gold and silver, respectively. A TI metasurface based on the MIM configuration was designed and its supported GSP resonance was identified through both mode analysis and strong magnetic field enhancement in the spacer layer. We optimized a brick structure to have a large cross-polarization conversion efficiency of up to 34% at 500 nm while the co-polarized light was simultaneously suppressed below 0.08%. By employing the principle of the geometric phase, a complete $2\pi$ phase coverage of nearly constant efficiency was achieved in the TI metasurfaces, leading to an efficient plasmonic platform for far-field wavefront manipulation in the UV-to-visible wavelength range. Furthermore, the large phase modulation of the GSP resonance was used to design a birefringent metasurface that was capable of directional surface plasmon polariton (SPP) excitation, depending on the polarization of the incident light.

## 2. Results and Discussion

The finite element software CST Studio Suite 2020 (Simulia, Johnston, RI, USA) with unit cell boundary conditions was used for all the modeling. The permittivity of $Bi_2Te_3$ adapted from [23] was used as a metal back-film and as nanoscatterers placed on an insulating spacer layer with nondispersive $SiO_2$ ($n = 1.5$). Air ($n = 1$) was chosen as the

surrounding medium. Only bulk TI was used for all simulations due to the neglectable influence of the surface in the studied spectral range [23]. Although this study only investigated TI metasurfaces numerically, the resulting structures shown are feasible for fabrications based on current nanofabrication technology. More specifically, our TI metasurfaces can be fabricated through electron-beam lithography on the TI thin films deposited via physical and chemical vapor deposition [14,35–37].

The most basic material requirement for plasmonic resonances is a negative real part $\varepsilon_1$ of the complex dielectric function $\varepsilon = \varepsilon_1 + i\varepsilon_2$. Interband transitions lead to a Lorentz-like permittivity with a peak in the $\varepsilon_2$ spectrum at the corresponding transition energy [21]. According to the Kramers–Kronig relation, linking the real and imaginary part of $\varepsilon$, a peak in $\varepsilon_2$ results in an anomalous jump in $\varepsilon_1$ [38]. If the oscillation amplitude of the interband transition is high enough, $\varepsilon_1$ has a high value at the lower energy side, while eventually turning negative for higher energies. This is exactly the case for $Bi_2Te_3$ (see Figure 1a). Due to a strong interband transition at around 780 nm ($\approx$1.6 eV), $\varepsilon_1$ reaches values above 50 on the long wavelength side, producing a material that is an outstanding candidate for dielectric Mie-like resonances with high mode confinement due to the large refractive index. On the small wavelength side, $\varepsilon_1$ becomes negative at around 700 nm. Around 500 nm, it reaches values below −15, revealing a pronounced metallic character that enables plasmonic resonances (marked in grey). This sets TI apart from most insulators, such as silicon, in which $\varepsilon_1$ remains positive for energies above the interband transition (see Figure A1c). Notably, the physical origin of the negative $\varepsilon_1$ in TIs is substantially different from that of noble metals since bulk TIs do not rely on free charge carriers. Therefore, $Bi_2Te_3$ does not have a plasma-cutoff frequency and $\varepsilon_1$ gradually approaches zero in the UV region. Note that $Bi_2Te_3$ has a lower $\varepsilon_1$ at optical frequencies than other TI materials, including $Bi_2Se_3$ (see Figure 2 in [23]), and therefore it was chosen for our metasurface designs.

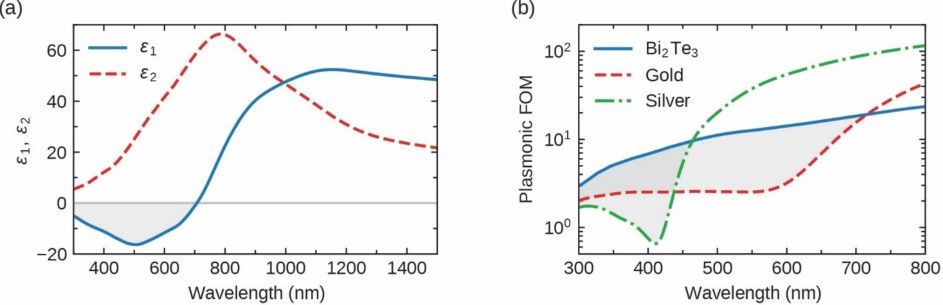

**Figure 1.** (**a**) Dielectric function of $Bi_2Te_3$ adapted from [23] with a large absorption peak for $\varepsilon_2$ around 770 nm. Following the Kramers–Kronig relation, $\varepsilon_1$ had an anomalous jump to negative values for wavelengths smaller than 700 nm, allowing plasmonic resonances. (**b**) Comparison of the plasmonic figure of merit (FOM) of $Bi_2Te_3$, gold, and silver [34], revealing superior plasmonic performance of $Bi_2Te_3$ for wavelengths below roughly 710 and 440 nm compared to gold and silver, respectively.

To investigate the capabilities of $Bi_2Te_3$ in contrast to traditionally used metals, Figure 1b shows the plasmonic FOM [39], which was calculated using

$$\mathrm{FOM} = \frac{\mathrm{Re}(k_{\mathrm{SPP}})}{2\pi\mathrm{Im}(k_{\mathrm{SPP}})},\tag{1}$$

with $k_{\mathrm{SPP}} = k_0\sqrt{\frac{\varepsilon_{\mathrm{metal}}\varepsilon_{\mathrm{air}}}{\varepsilon_{\mathrm{metal}}+\varepsilon_{\mathrm{air}}}}$ as the surface plasmon wavevector between the presented materials and air. Intuitively, it can be understood as the number of wavelengths a plasmon polariton can travel until its energy drops below $1/e$. The permittivity values of gold and silver were adapted from [34]. For all three materials, the FOM significantly decreased toward smaller wavelengths. Remarkably, $Bi_2Te_3$ surpassed gold and silver below roughly 710 and 440 nm, respectively. Marked in grey are the spectral regions in which $Bi_2Te_3$ outperformed gold and silver in terms of their plasmonic capabilities. This revealed a large

spectral range in which $Bi_2Te_3$ and TIs in general are better alternatives to noble metals. The operational wavelength range of our metasurface designs was chosen to be in this region of a superior plasmonic FOM.

### 2.1. TI-Based GSP Metasurface

To confirm the presence of plasmonic resonances in structured MIM metasurfaces, a well understood nanobrick design [40] was used. A squared brick with length $L = 110$ nm and height $h_b = 20$ nm was placed on top of an $h_f = 100$ nm $Bi_2Te_3$ film and an $h_s = 50$ nm $SiO_2$ spacer layer (see Figure 2a). To identify different types of resonances, the pitch $\Lambda$ between bricks was varied from 160 to 280 nm. Figure 2b shows the corresponding normalized reflectance over a large spectral range from the near UV to the near IR as a 2D color plot for plane-wave excitations along the $x$-direction. Three distinct modes are apparent as local minima in the reflectance spectrum. The dashed line at $\Lambda = 220$ nm indicates the single spectral line plotted below. Here, the three minima are numbered and the corresponding $x$- or $y$-components of the electric and magnetic fields are shown in Figure 2c,d, respectively. The cutting plane was in the middle of the unit cell and the color-code was normalized to the incident field. In Figure 2b, the mode at the small wavelength side labeled 1 in Figure 2c shows magnetic fields of comparable strength on the top of and below the brick, while the electric field indicates a delocalized resonant behavior due to relatively high fields between the structures [41]. The first-order plasmonic grating coupling condition, marked in blue in Figure 2b, was fulfilled when the pitch was equal to $\lambda_{SPP}$ at the metal–insulator interface [42]: $\Lambda = \lambda_{SPP} = \lambda_0 \sqrt{\frac{\varepsilon_{Bi_2Te_3} + \varepsilon_{SiO_2}}{\varepsilon_{Bi_2Te_3} \varepsilon_{SiO_2}}}$. The small offset was mainly due to the simplified formula, which only accounts for a $Bi_2Te_3$–$SiO_2$ interface and not for a more complex $Bi_2Te_3$–$SiO_2$–air interface [43]. In the mode profile of the second dip, the enhanced magnetic fields below the nanobrick indicate a GSP mode. The asymmetric field below and above the brick forming a magnetic resonance in combination with the localized shape of the electric field is typical for the lowest order GSP [41]. Above roughly 700 nm, the real part of the dielectric function $\varepsilon_1$ was positive (as already shown in Figure 1a), leading to the dielectric behavior of $Bi_2Te_3$. The first order Fabry–Perot mode for a 100-nm-thick $Bi_2Te_3$ film calculated using $\lambda_{FP} = \frac{1}{2} h_f n_{Bi_2Te_3}$ is marked in green. This vertical line matches the third reflectance dip quite well. As expected from a Fabry–Perot mode, its spectral position was nearly independent of the pitch. The corresponding mode profile in Figure 2d shows magnetic fields deeply penetrating the $Bi_2Te_3$ back-film and fields larger than zero beneath it underlining the dielectric behavior. Being able to excite plasmonic and dielectric modes inside the same structure, depending on the wavelength, highlights the versatility of TIs for nanophotonic applications.

### 2.2. Far-Field Functionalities

So far, the potential of $Bi_2Te_3$ for GSP metasurfaces has been presented and now their performance in applications is investigated. Engineering the far-field response of metasurfaces has become one of the most promising fields in nanophotonics [44] for beam steering [45], flat lenses [29], and holograms [46]. Therefore, there is a high demand for functional metasurfaces that can outperform conventional bulky optical elements.

By introducing phase discontinuities at an interface in the form of nanoscatterers, Snell's law can be expanded with an additional term [1], yielding:

$$\sin \theta_r n_i - \sin \theta_i n_i = \frac{\lambda_0}{2\pi} \frac{d\phi}{dx}, \tag{2}$$

where $\theta_i$ and $\theta_r$ are the incidence and reflectance angles, $\lambda_0$ is the wavelength of the incident light, and $n_i$ is the refractive index of the initial media. $\frac{d\phi}{dx}$ denotes the position-dependent phase gradient at an interface. As such, ultrathin metasurfaces with different photonic functionalities, such as beam steering, diffraction-limited light focusing, and holographic displays, have been achieved from imprinting the phase gradients of a linear grating [45],

a parabolic lens [29], and a hologram [30], respectively. However, most nanoscatterers are not capable of achieving a full $2\pi$ phase coverage. Simple dipole resonances, e.g., in a plasmonic nanorod, only cover a phase range of $\pi$ when the excitation wavelength is tuned over its resonance wavelength [1]. However, to date, the development of a TI metasurface covering the full $2\pi$ phase coverage for far-field wavefront manipulation is still elusive. Here, we demonstrate the design of a highly efficient $Bi_2Te_3$ metasurface based on the principle of GSPs, leading to the full $2\pi$ phase coverage that is essential for efficient far-field beam engineering in the UV and visible light ranges.

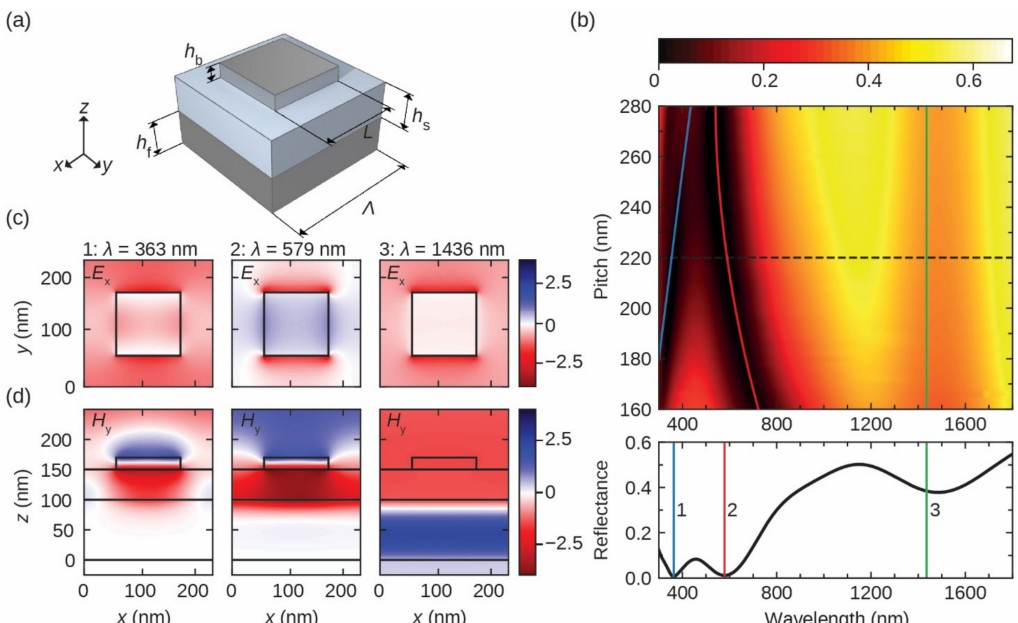

**Figure 2.** (**a**) Unit cell sketch of a gap–surface plasmon metasurface consisting of a quadratic $L$ = 110-nm-long and $h_b$ = 20-nm-high $Bi_2Te_3$ brick, an $h_s$ = 50 nm $SiO_2$ spacer layer, and an $h_f$ = 100-nm-thick $Bi_2Te_3$ back-film. The pitch $\Lambda$ was varied from 160 to 280 nm. (**b**) The corresponding reflectance spectra with the color code referring to the normalized reflectance. Three distinct modes are visible in the spectra, with each one marked with a solid line. The blue line corresponds to the (1,0) surface plasmon polariton (SPP) mode, which redshifted with the pitch following the plasmonic grating coupling condition [42]. The gap–surface plasmon mode and the dielectric Fabry–Perot mode inside of the $Bi_2Te_3$ back-film is marked in red and green, respectively. The black dashed line at 220 nm in (**b**) indicates the single spectral line plotted below. All three reflectance dips are marked with lines and labeled with numbers from 1–3, which correspond to the electric and magnetic field profiles shown in (**c,d**), respectively. The $x$-component of the electric field is plotted in the $x$-$y$ plane and the $y$-component of the magnetic field is plotted in the $x$-$z$ plane. The color codes were normalized to the incident $x$-polarized field.

To realize the full $2\pi$ phase coverage, we present the design of a TI GSP metasurface based on the principle of the Pancharatnam–Berry (PB) phase [47]. So far, PB-phase-based GSP metasurfaces have already been successfully fabricated, yielding conversion efficiencies around 80% in the visible and near IR [30] and above 90% in the microwave range [48]. Here, we showed that a $Bi_2Te_3$-based GSP metasurface, operating in the visible light range, is capable of arbitrarily tuning the phase of the reflected cross-polarized light. By rotating the birefringent nanobricks between 0° and 180°, a $2\pi$ phase coverage was achieved [49].

The cross-polarization efficiency was optimized by sweeping the structural parameters (spacer layer height, brick height, length, width, and pitch distance) of the nanobrick with the consideration of fabrication feasibility. The cross-polarization efficiency was defined as the square of half the difference between the S-parameters for the polarization along the

long and short transverse axes [49]. The final design for the metasurface had a nanobrick length of 170 nm, a width of 45 nm, a height of 45 nm, a 50 nm spacer layer, and a 200 nm pitch distance. It yielded a conversion efficiency as high as 34% (meaning a cross-polarization reflectivity of 0.34) at 500 nm (see Figure 3a). Furthermore, the conversion efficiency was above 20% over nearly the entire visible range (400 to 800 nm). The co-polarized light was suppressed below 0.08% at 500 nm and below 6% for the entire visible spectrum. Since the backplane of the MIM structure was optically dense, the transmittance was neglected, hence the absorbance $A$ was given by $A = 1 - R$, with $R$ as the sum of the reflected cross- and co-polarized light (see Figure A2).

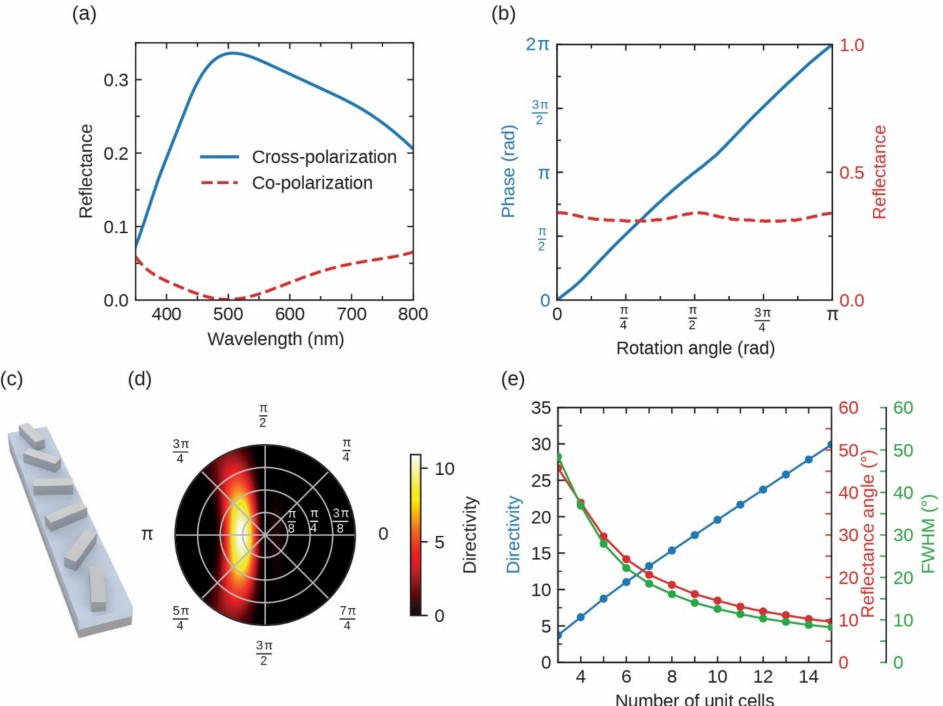

**Figure 3.** In all the figures (**a**–**e**), optimized $Bi_2Te_3$ nanobricks with a length of 170 nm, a width of 45 nm, and a height of 45 nm, with a 200 nm pitch and a 50 nm $SiO_2$ spacer layer were used. (**a**) shows the reflectance spectra for co- and cross-polarized light for excitation with circular light with a conversion efficiency above 20% over nearly the entire visible spectrum and a maximum of 34% at 500 nm. In (**b**), the phase response of the nanobrick is shown for different rotational angles, achieving a $2\pi$ phase coverage while the reflectance stayed nearly constant. In (**c**), a sketch of the supercells used in (**d**) is shown, which consisted of six nanobricks anomalously reflecting light at 24.3°, which is shown in the polar color plot in (**d**). ϕ is the angle in the *x-y* plane and represents the angular component ranging from 0 to $2\pi$, while the *z*-angle Θ is the radial component forming a hemispherical surface. A pronounced peak of a directivity of 11.0 is visible at Θ = 24.3°. The connection between the number of unit cells in each supercell and the parameter of the anomalously reflected beam is given in (**e**). FWHM: full width at half maximum.

Due to the rather small feature sizes of our designed TI metasurface, fabrication imperfections must be considered. Variations in the height of the different layers can be neglected since film deposition either from chemical vapor deposition or atomic layer deposition has a very high accuracy [14]. The length and width of the nanobricks were set using lithography patterning, which might result in larger deviations. Figure A3 shows the cross- and co-polarized reflectance for the structure used in Figure 3a and for 10 nm detuned lengths and widths. Due to its smaller size, the detuning of the width had a higher impact on the efficiency than the length. Here, a 10 nm reduced width yielded the lowest conversion efficiency, 3% smaller at 500 nm than the optimized one. This indicated a robust metasurface design with a high tolerance against fabrication imperfections. Note that the 180 nm long brick yielded a

higher cross-polarization efficiency than the 170-nm-long one. However, the latter was used in Figure 3 with sufficiently large gaps between the bricks to ensure fabrication feasibility.

In Figure 3b, the phase response is plotted against the rotation angle at 500 nm for the same structure, confirming a linear $2\pi$ phase coverage that was achieved from the rotation of nanobricks. To demonstrate the beam-steering capabilities, a metasurface with a linear phase gradient was designed (see Figure 3c). A supercell consisting of six bricks with different phase responses covering the full $2\pi$ range was simulated. The combined length was equal to six times the unit cell used in Figure 3a,b. The angle-dependent far-field response is shown in Figure 3d. $\phi$ is the angle in the *x-y* plane and represents the angular component, while the *z*-angle $\Theta$ is the radial component forming a hemispherical surface. The far-field response is given in the unitless directivity, which denotes the intensity normalized to a Huygens scatterer with unidirectional scattering. For a phase gradient of $2\pi$ over six 200 nm unit cells, the generalized Snell's law predicts a reflectance angle of $24.6°$, which matches well with the maximum in Figure 3d at $24.3°$. Furthermore, a main lobe with a directivity of 10.0, combined with no apparent side lobes, provided a very clean signal. Similar simulations were performed with different numbers of unit cells that imprint an overall $2\pi$ phase grating. The simulated reflectance angles are shown in red in Figure 3e. The smaller the grating pitch (the number of 200 nm unit cells), the larger the anomalous reflectance angle. However, reducing the number of unit cells increased the phase-step gap in the discretization of a linear grating phase, eventually reducing the directivity maximum (in blue in Figure 3e), and the main lobe broadened with an increased full width at half maximum (FWHM) (in green). Therefore, depending on the application, one must choose between large anomalous reflectance angles with a high phase gradient and high lobes with a smooth phase gradient.

### 2.3. Near-Field Functionalities

Another intriguing feature of GSP metasurfaces is their extraordinary capabilities of launching SPPs. Due to their continuous metal back-film, highly efficient SPP coupling can be achieved if the length of a $2\pi$ phase gradient matches the surface plasmon wavelength $\lambda_{SPP}$ [32,50]. In 2014, Pors et al. [32] showed that unidirectional polarization controlled SPP coupling at telecommunication wavelengths. By designing a birefringent GSP metasurface, they controlled the directional excitation of SPPs depending on the polarization of the incident light. In the following, we demonstrate the design of a birefringent GSP metasurface for $Bi_2Te_3$ TI films at optical frequencies. The optimized wavelength was again 500 nm. To achieve polarization-dependent directional SPP excitation, a birefringent supercell with two independent phase gradients for *x*- and *y*-polarized light and side lengths equal to $\lambda_{SPP}$ was designed. Since at least three different phase responses are needed to create a blazed phase grating, a $3 \times 3$ nanobrick supercell design was chosen (see Figure 4a). Using more nanobricks smoothens the phase gradient but at the cost of fabrication feasibility. In Figure 4b, the length and width of the nanobricks were tuned from 30 to 150 nm for a unit cell size of 160 nm, which is roughly one-third of $\lambda_{SPP}$. The color code indicates the phase response for *x*-polarization. In Figure A4, the corresponding reflectance is shown. The plot can be divided into two regions in which the phase behaved completely differently. For widths below 55 nm, the system was overdamped, while it was underdamped at larger widths. Using coupled-mode theory and a one-port resonator model, this effect is explained in great detail in [51–53]. At the transition line, the system is critically damped. Above it, radiation loss is dominant, while below it, intrinsic loss dominated. This led to an abrupt phase jump at a width of 55 nm for lengths above the GSP resonance. Furthermore, the phase gradient for different lengths has opposite signs in the two regions. Figure A5 shows the different phase behaviors for under- and overdamped systems around a 500 nm excitation. In the bottom part of Figure 4b, it is negative, covering a range of $\pi$, while in the upper part, it is positive, covering more than $\pi$, and by including even larger structures, a $2\pi$ phase difference should be possible. To keep the SPP excitation efficiency high, only structures in the underdamped regime where radiation loss dominated were chosen.

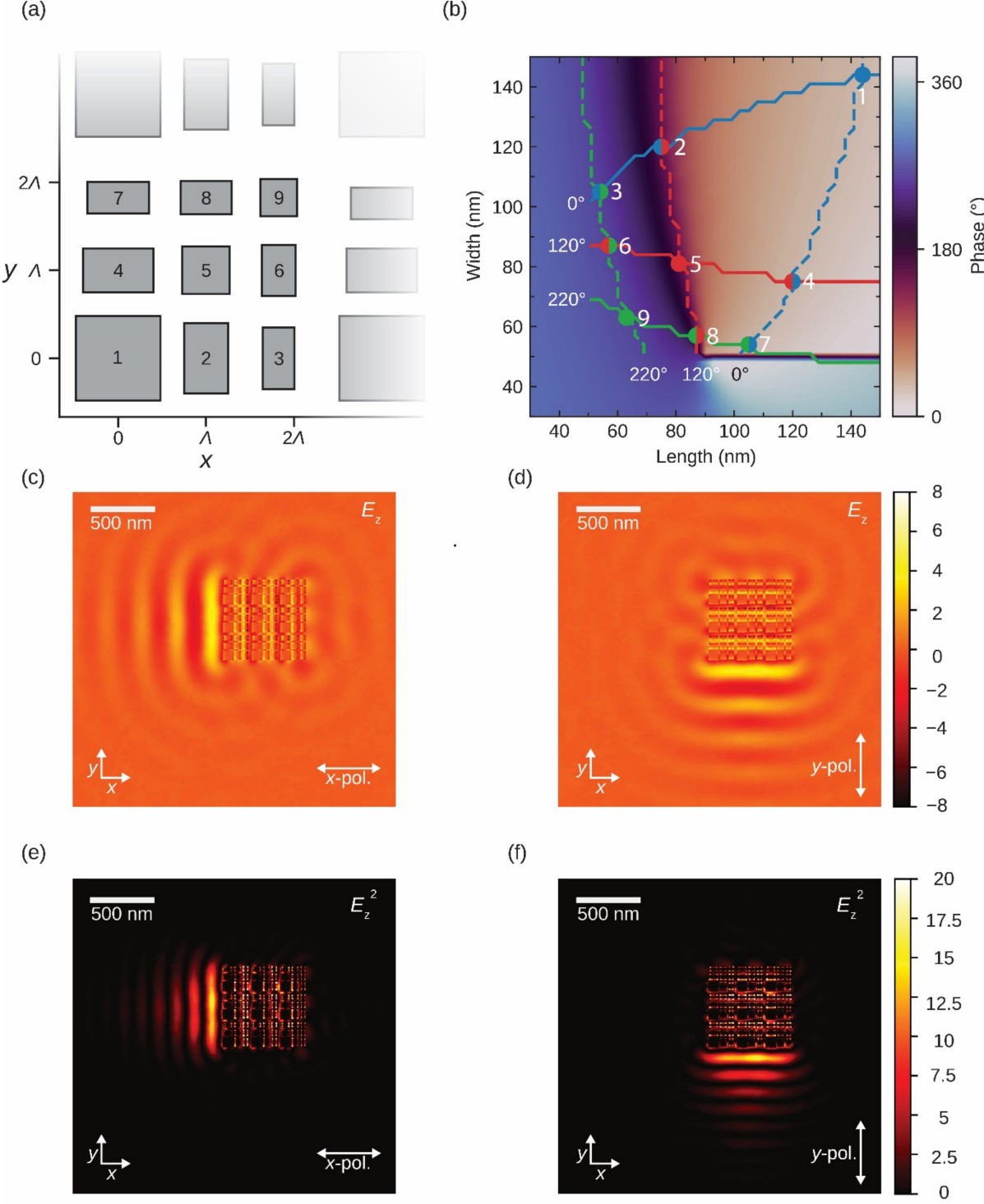

**Figure 4.** (**a**) Schematics of a birefringent metasurface that was capable of exciting directional SPPs. The supercell consisted of nine nanobricks in a 450 × 450 nm area with an average pitch of Λ. In (**b**), the length and width of a $Bi_2Te_3$ nanobrick with a unit cell size of 160 nm and a spacer layer of 30 nm was tuned from 30 to 150 nm, where the color code shows the phase response for *x*-polarized light. At a 55 nm width, a transition from an overdamped to an underdamped system was visible. The isolines at 0°, 120°, and 220° for *x*-polarization in the underdamped region are marked with solid lines, while the isolines for the *y*-polarization are marked with dashed lines. The nine intersection points marked with dots each correspond to one of the bricks in (**a**). Note that unequal phase steps were chosen to create the 2π phase gradient over the supercell length. To maintain a linear gradient, bricks with 220° phase responses were slightly shifted. (**c**,**d**) show the *z*-component of the electric field for *x*- and *y*-polarized light, respectively, while (**e**,**f**) show the corresponding intensities. (**c**–**f**) were normalized to the excitation field and reveal highly directional SPP excitation.

For equal phase steps in a $3 \times 3$ supercell, one would typically use relative phase responses of 0°, 120°, and 240°. However, due to the small unit cell size used here, the structures can only be quite small. This makes it particularly challenging to achieve a 240° phase difference for one polarization while keeping the phase for the other polarization constant. Therefore, slightly uneven phase steps were chosen at 0°, 110°, and 220°. The isolines of these values are labeled in Figure 4b for *x*- (solid) and *y*- (dashed) polarization. The crossing points mark the length and width of the nine nanobricks used in the supercell pictured in Figure 4a. The structures are not equally spaced within the supercell. Their positions correspond to the phase shifts they induced to achieve a linear phase gradient. This led to a larger pitch between the first and third row and a smaller pitch between the second and third row. However, due to the differently sized bricks, the smallest gap was still between the first and second row. To illustrate the unidirectional polarization-dependent SPP coupling, a $3 \times 3$ supercell array on a continuous $Bi_2Te_3$ film was simulated under plane wave excitation. The *z*-component of the electric field and the corresponding intensity distributions for *x*- and *y*-polarized light are shown in Figure 4c–f, respectively. The exact calculation of the SPP wavelength within the metasurface was challenging due to the presence of the 30 nm spacer layer, in addition to the nanobricks. Thus, the supercell length was iteratively changed to a final length of 450 nm until the coupling was maximized. However, the smallest feature size was still larger than the 40 nm available to standard fabrication methods. Clearly, SPPs were predominantly excited into one direction following the designed phase gradient. This emphasizes the high potential of TIs for near-field metasurfaces.

## 3. Conclusions

In this work, we numerically investigated the capabilities of TI GSP metasurfaces for applications in the visible spectral range. Due to strong interband absorption in the visible spectrum resulting in a negative $\varepsilon_1$, the used $Bi_2Te_3$ TI had a higher plasmonic FOM than conventional plasmonic materials, including gold and silver at wavelengths smaller than 710 nm and 440 nm, respectively. We demonstrated the excitation of GSP resonances, as well as SPP resonances, within a $Bi_2Te_3$ nanobrick MIM metasurface using spectral and mode analysis. Furthermore, the beam-steering capabilities of the TI GSP metasurfaces were demonstrated by using birefringent nanobricks and the geometric phase principle. A cross-polarization efficiency of 34% at 500 nm and above 20% in the whole visible spectrum was achieved. Simultaneously, the co-polarized light was suppressed below 0.08% at 500 nm and below 6% in the visible spectrum. The beam shape and directivity in connection with the reflectance angle of the anomalously reflected beam were studied and showed a pronounced main lobe and a directivity above 30 for small angles. Our numerical study paves the way for future far-field functionalized TI-based GSP metasurfaces, such as holograms or lenses. Finally, we demonstrated a directional polarization-dependent SPP coupler using a birefringent TI GSP metasurface. Owing to their outstanding plasmonic performance in the visible and near UV, their CMOS compatibility, and their high thermal stability, TIs are a highly promising new material class for a large variety of plasmonic applications.

**Author Contributions:** H.R. and A.A. proposed the idea and conceived the simulation; A.A. performed the numerical simulations of the TI-based GSP metasurfaces; A.A., S.A.M. and H.R. contributed to the data analysis. All authors have read and agreed to the published version of the manuscript.

**Funding:** H.R. acknowledges the funding support from the Humboldt Research Fellowship from the Alexander von Humboldt Foundation. H.R. acknowledges the funding support from the Macquarie University Research Fellowship (MQRF) from Macquarie University. S.A.M. acknowledges the funding support from the Deutsche Forschungsgemeinschaft, the EPSRC (Engineering and Physical Sciences Research Council) (EP/M013812/1), and the Lee-Lucas Chair in Physics.

**Informed Consent Statement:** Not applicable.

**Data Availability Statement:** The data that support the findings of this study are available from the corresponding author upon reasonable request.

**Conflicts of Interest:** The authors declare no conflict of interest.

## Appendix A

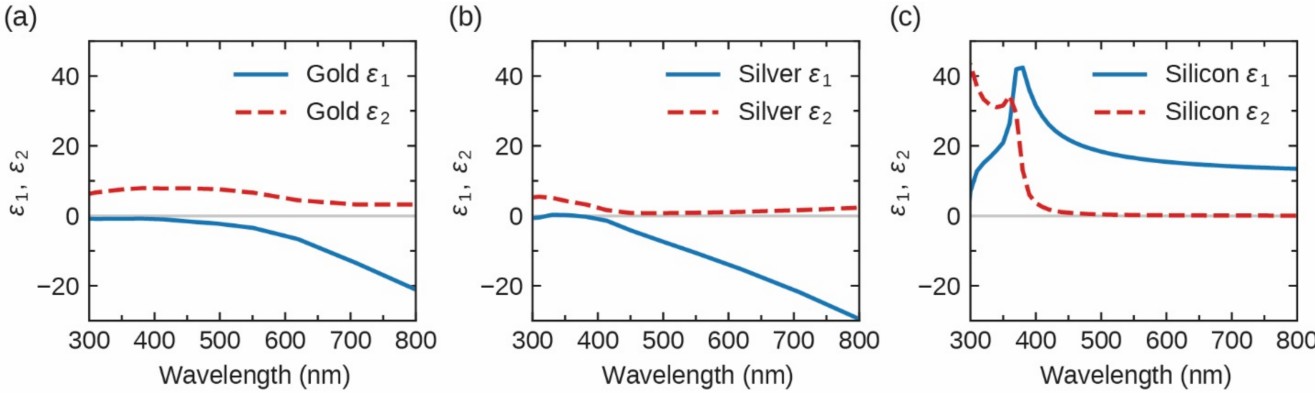

**Figure A1.** Permittivity of the used gold (**a**) and silver (**b**) adapted from [34]. Permittivity of Si (**c**) adapted from [54] with an interband absorption around 280 nm. Due to the smaller oscillation amplitude compared to $Bi_2Te_3$, $\varepsilon_1$ did not become negative.

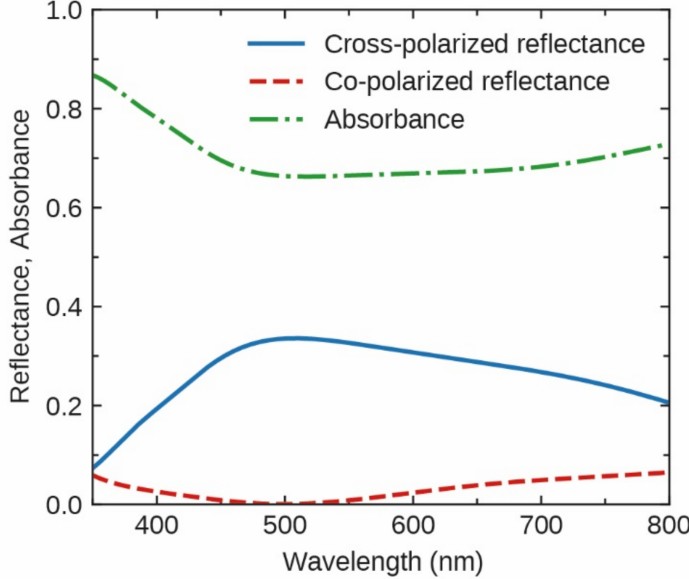

**Figure A2.** Cross- and co-polarized reflectance amplitudes ($R_{\text{cross}}$, $R_{\text{co}}$) and absorbance $A$ of the $Bi_2Te_3$ nanobrick metasurface used in Figure 3a with a 170 nm length, 45 nm width, 45 nm height, 200 nm pitch, and 50 nm $SiO_2$ spacer layer. Since the transmittance was equal to zero, the absorbance was calculated using $A = 1 - R = 1 - R_{\text{cross}} - R_{\text{co}}$.

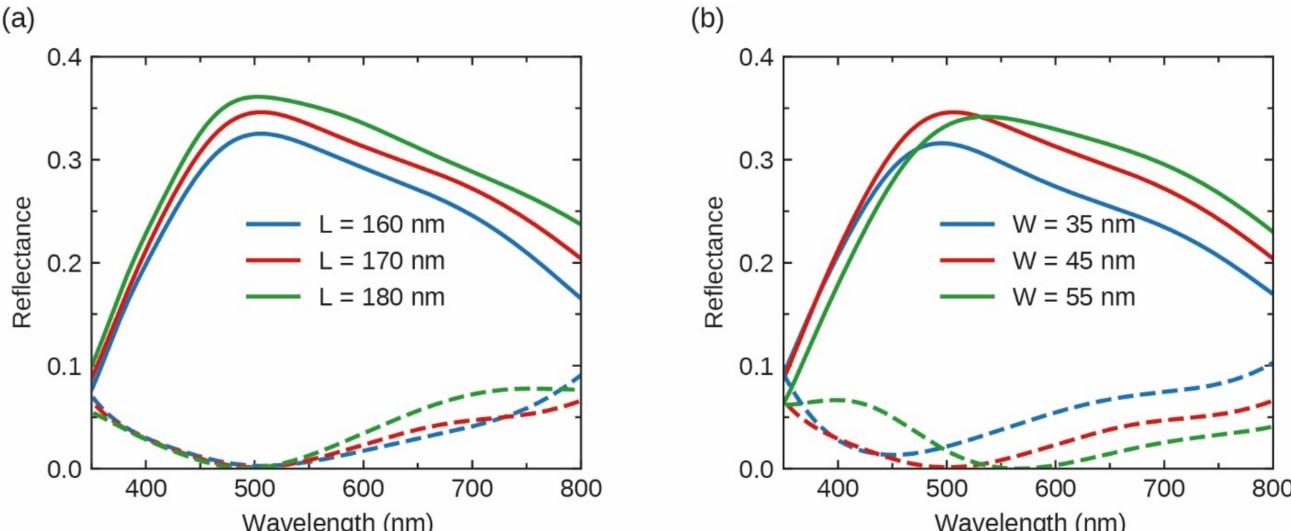

**Figure A3.** In (**a**), the length *L* and in (**b**) the width *W* of the nanobrick metasurface used in Figure 3a was detuned by ±10 nm to test the stability against fabrication imperfections. For the original and detuned structures, the cross- and co-polarized reflectance amplitude is indicated with solid and dashed lines, respectively. In (**a**), the efficiency increased with the nanobrick length; however, longer bricks led to smaller gaps between them. To keep a balance between efficiency and fabrication feasibility, a length of 170 nm was chosen for the final design. The cross-polarization efficiency at 500 nm was decreased by 2% for a 10 nm smaller length. Furthermore, it was the highest for the chosen width of 45 nm and the lowest with 3% less for a 35 nm width (**b**).

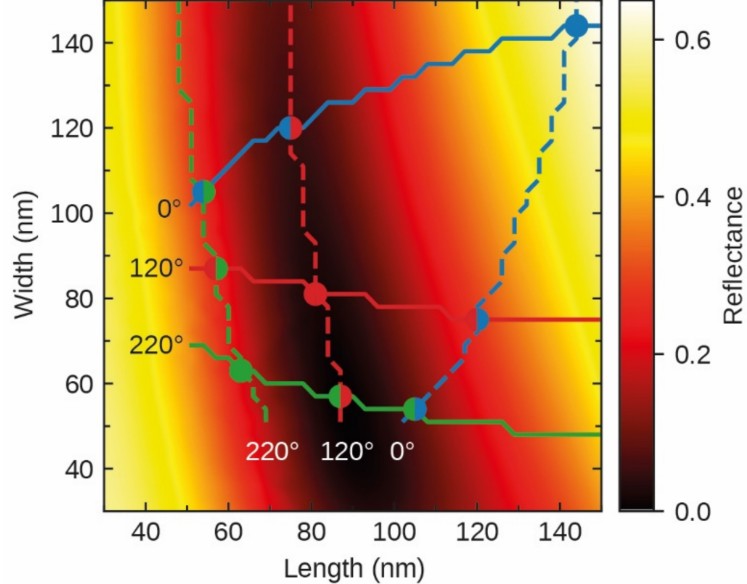

**Figure A4.** Reflectance amplitude at 500 nm in color for different nanobrick widths and lengths. The phase response isolines for x- and y-polarized light at 0, 120°, and 220° degrees are marked with solid and dashed lines, respectively. The parameters of the bricks at the crossing points marked with dots were used for the supercell design.

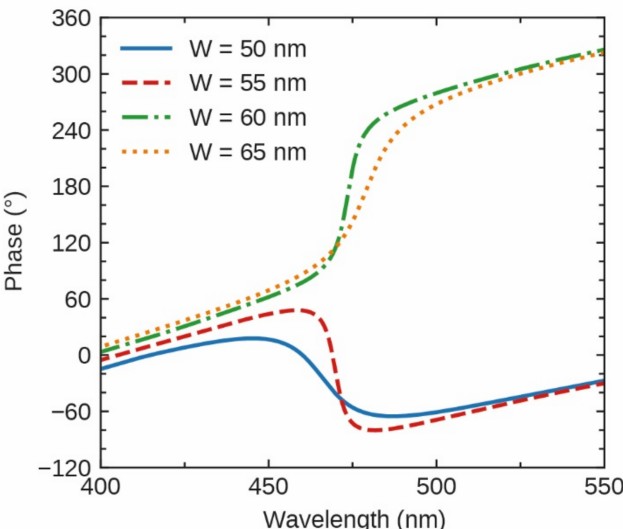

**Figure A5.** Phase behavior of the nanobricks with a 30 nm spacer layer, 30 nm brick height, 160 nm pitch, 80 nm length, and widths of 50, 55, 60, and 65 nm. A transition from an overdamped to an underdamped system is indicated by the fast nearly 2π phase jump between a width of 55 and 60 nm.

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
