# Peer review of "Topological-Insulator-Based Gap-Surface Plasmon Metasurfaces"

_photonics, doi:10.3390/photonics8020040_

Round 1

Reviewer 1 Report

The manuscript under review is dedicated to the numerical investigation of gap-plasmon metasurfaces made of a topological insulator (TI) material. The paper is well written, interesting, and suitable for the Photonics journal. In my opinion, it can be accepted for publication after a minor revision according to the following comments.

1. The authors mentioned that the parameters of the metasurface (brick length, width, and height) used for obtaining efficient polarization conversion were found using numerical optimization. Could the authors give some details on the optimization process?
2. It would be useful to add the absorption and transmittance plots to Fig. 3(a), in order to make it clear where the non-reflected part of the energy goes.
3. Could the authors briefly comment on the fabrication techniques that can be used to implement the TI metasurfaces?

Author Response

The manuscript under review is dedicated to the numerical investigation of gap-plasmon metasurfaces made of a topological insulator (TI) material. The paper is well written, interesting, and suitable for the Photonics journal. In my opinion, it can be accepted for publication after a minor revision according to the following comments.

Reply 1: We thank the reviewer for the positive confirmation of the quality of our work and the recommendation of publishing our manuscript in Photonics.

  1. The authors mentioned that the parameters of the metasurface (brick length, width, and height) used for obtaining efficient polarization conversion were found using numerical optimization. Could the authors give some details on the optimization process?

Reply 2: We thank the reviewer for raising this point. The optimized structure was found via tuning of all the structural parameters, including thespacer layer height, brick height, length, width, and pitch distance. The optical responses of simulated structures were numerically evaluated through cross-polarization efficiency at different wavelengths, based on the Jones matrix method [50]. The numerical nature of the technique was pointed out to highlight the method (finite element simulations) we used to obtain these parameters. To clarify this optimization, we further expanded the description:

Here we show that a Bi2Te3-based GSP metasurface, operating in the visible is capable of arbitrarily tuning the phase of the reflected cross-polarized light. By rotating the birefrin-gent nanobricks between 0° and 180° a 2π phase coverage is achieved [50].
The cross-polarization efficiency was optimized by sweeping structural parameters (spacer layer height, brick height, length, width, and pitch distance) of the nanobrick with the consideration of fabrication feasibility. The cross-polarization efficiency was defined as the square of half the difference between the S-parameters for the polarization along the long and short transverse axes [50]. The final design for the metasurface has a nanobrick length of 170nm, a width of 45nm, and a height of 45nm, a 50nm spacer layer and a 200nm pitch distance. It yields a conversion efficiency as high as 34% (meaning a cross-polarization reflectivity of 0.34) at 500nm, see Fig. 3a.

  1. It would be useful to add the absorption and transmittance plots to Fig. 3(a), in order to make it clear where the non-reflected part of the energy goes.

Reply 3: Since the Bi2Te3 back-film of the MIM metasurface used in Fig. 3a is optically dense in the investigated spectral region, the transmittance is neglectable. To emphasize this, we added the following sentence and Fig. S2.

Since the backplane of the MIM structure is optically dense, the transmittance is neglected hence the absorbance A is given by A=1-R with R as the sum of the reflected cross- and co-polarized light, see Fig. S2.

Figure S2. Cross- and co-polarized reflectance amplitude (Rcross, Rco) and absorbance A of the Bi2Te3 nanobrick metasurface used in Fig. 3a with 170nm length, 45nm width, 45nm height, 200nm pitch, and 50nm SiO2 spacer layer. Since the transmittance is equal to zero, the absorbance is calculated by .

  1. Could the authors briefly comment on the fabrication techniques that can be used to implement the TI metasurfaces?

Reply 4: We thank the reviewer for the comment and added our brief comments on the potential fabrication techniques to fabricate our TI metasurfaces.

Although this paper only investigates TI metasurfaces numerically, the resulting struc-tures shown are feasible for fabrication based on current nanofabrication technology. More specifically, our TI metasurfaces can be fabricated through electron-beam lithography on the TI thin films deposited via physical and chemical vapor deposition [17][36]–[38].

Reviewer 2 Report

The paper by Aigner et al. reports interesting results on topological insulator-based gap-plasmon metasurfaces. The degree of interest is very good, as well as the potential impact. However, some amendments are necessary before publication.

1) The authors have not considered that plasmonic excitations in bismuth chalcogenides are particularly relevant for plasma-wave Terahertz photodetectors. 

2) Review papers on plasmonics and photonics with topological insulators should be cited.

3) The Dirac plasmon of topological insulators hybridizes with surface plasmon. This aspect should be briefly mentioned with appropriate discussion.

4) Dirac plasmon should be introduced with a more complete overview of the state of the art. In the current version of the manuscript, only "historical" information is present, but without utility for readers who instead need scientific information.

4) Bi2Se3 is largely more studied than Bi2Te3 and Sb2Te3 as 3D topological insulator. A comparison of their results with the case of Bi2Se3 is required

Author Response

The paper by Aigner et al. reports interesting results on topological insulator-based gap-plasmon metasurfaces. The degree of interest is very good, as well as the potential impact. However, some amendments are necessary before publication.

Reply 1: We thank the reviewer for the positive confirmation of the quality and potential impact of our work and the recommendation of publishing our manuscript in Photonics after some amendments. We have gratefully adopted the suggestions for improving and extending the introductory part about the fundamentals and the-state-of-the-art of Dirac plasmons in TIs. We have added one section in the paper (see Reply 5 and paragraph 2-3 on page 1-2 in the manuscript) to address the comments.

1) The authors have not considered that plasmonic excitations in bismuth chalcogenides are particularly relevant for plasma-wave Terahertz photodetectors. 

Reply 2: We have added the discussion about plasma-wave terahertz photodetectors in paragraph 2 page 2, which is an exciting platform for future photodetectors in the terahertz range, and two different design approaches have been mentioned:

Particularly interesting applications for TIs are THz photodetectors, which utilize plasma-wave oscillations in metal-TI-metal heterostructures [18] and field effect transistors [19].

2) Review papers on plasmonics and photonics with topological insulators should be cited. 

Reply 3: Several review papers about TI plasmons and Dirac plasmons have been added::

Stauber, Tobias. "Plasmonics in Dirac systems: from graphene to topological insulators." Journal of Physics: Condensed Matter (2014)

Ginley, T., et al. "Dirac plasmons and beyond: the past, present, and future of plasmonics in 3D topological insulators." MRS Communications (2018)

Lai, Yi-Ping, et al. "Plasmonics in topological insulators." Nanomaterials and Nanotechnology (2014)

Politano Antonio, Leonardo Viti, and Miriam S. Vitiello. "Optoelectronic devices, plasmonics, and photonics with topological insulators." APL Materials (2017)

3) The Dirac plasmon of topological insulators hybridizes with surface plasmon. This aspect should be briefly mentioned with appropriate discussion.

Reply 4: We added some text (see the numbered sections below) to explain the interaction of Dirac electrons with non-topological electrons (e.g., bulk electrons), which forms combined plasmons (1). Furthermore, we discussed the hybridization of Dirac plasmons of the top and bottom surface of thin TI films, which forms acoustic and optical modes (2). Finally, we also mentioned the interaction of Dirac plasmons with phonons and the resulting the Fano-like line shape (3).

1: It is worth mentioning that not all charge carriers involved in a Dirac plasmon are massless Dirac electrons. In fact, it is a combination of Dirac electrons and massive electrons constituted from a non-topological 2D electron gas and bulk electrons [16], [21], [22]. This effect can be exploited by photo-doping TI gratings for ultra-high modulation depths above 2,400% as demonstrated by Sangwan Sim, et al. [22].

2: Due to the subwavelength thickness of most studied TIs in the THz region, plasmons are excited at the top and bottom layer simultaneously, yielding a coupled system with acoustic (dark) and optical (bright) plasmon modes which are purely spin- and charge-like, respectively [13]–[16].

3: The broad plasmon modes can interact with narrow phonon modes and result in the Fano-type line shapes [17][18].

4) Dirac plasmon should be introduced with a more complete overview of the state of the art. In the current version of the manuscript, only "historical" information is present, but without utility for readers who instead need scientific information.

Reply 5: Since our metasurface was designed for optical frequencies and uses relatively thick TI films, the contribution of the conducting surface states to the overall optical response and permittivity is small. In this spectral region the bulk acts as a metal hence the overwhelming majority of the charge carriers involved on the plasmonic resonances are bulk electrons and no Dirac plasmon-like behaviour will be present. For this reason, we initially had kept the text about Dirac plasmons short. However, to better illustrate the big picture and to give a more complete introduction to the topic, we added the following paragraph dedicated to introducing Dirac plasmons(paragraphs 2-3 Page 1-2):

They were introduced to the field of plasmonics in 2013 by the observation of temperature robust Dirac plasmons in TI gratings in the THz range [11]. By varying the grating width and film thickness several groups [12][10][13] were able to map the dispersion relation of the investigated plasmons confirming its topological nature. Due to the subwavelength thickness of most studied TIs in the THz region, plasmons are excited at the top and bottom layer simultaneously yielding a coupled system with acoustic (dark) and optical (bright) plasmon modes which are purely spin- and charge-like, respectively [13]–[16]. The broad plasmon modes can interact with narrow phonon modes and result in the Fano-type line shapes [17][18]. Dirac plasmons show outstanding long lifetimes, high tunability in the THz range, temperature stability up to room temperature, and a large mode index compared to traditional metals [12]. Particularly interesting applications for TIs are THz photodetectors, which utilize plasma-wave oscillations in metal-TI-metal heterostructures [19] and field effect transistors [20].
It is worth mentioning that not all charge carriers involved in a Dirac plasmon are massless Dirac electrons. In fact, it is a combination of Dirac electrons and massive electrons constituted from a non-topological 2D electron gas and bulk electrons [16], [21], [22]. This effect can be exploited by photo-doping TI gratings for ultra-high modulation depths above 2,400% as demonstrated by Sangwan Sim, et al. [22]. Typically, Dirac plasmons are studied in the THz regime, where the bulk acts as dielectric with relatively low loss.

4) Bi2Se3 is largely more studied than Bi2Te3 and Sb2Te3 as 3D topological insulator. A comparison of their results with the case of Bi2Se3 is required

Reply 6: We thank the reviewer for drawing attention to why we chose Bi2Te3 over Bi2Se3 and other TIs. In the reference “Plasmonics of topological insulators at optical frequencies” by Jun Yin, et al., the authors made an extensive comparison between the permittivities of TIs at optical frequencies, revealing the much more metallic character of Bi2Te3 compared to Bi2Se3 due to its higher negative real part of the permittivity (ε1) at around 500nm. Furthermore, ε1 of Bi2Te3 turns negative at around 700nm, allowing plasmonic excitation in the bulk nearly in the entire visible spectrum. In contrast, ε1 of Bi2Se3 turns negative at around 500nm, prohibiting plasmonic resonances in the bulk at larger wavelengths. In order to back up our selection of TI material more clearly, we added the following sentence:

Note that Bi2Te3 has a lower ε1 at optical frequencies than other TI materials including Bi2Se3 (see [26] Fig. 2) and therefore it was chosen for our metasurface designs.

Reviewer 3 Report

The authors design and numerically simulate topological insulator-based gap-plasmon metasurfaces as a new material class for plasmonic applications as holograms, lenses, and other devices.
They use chalcogenide materials to achieve higher figures of merit in visible and UV bands than metals in a metal-insulator-metal nanobrick metasurface.
By employing the PB phase principle, they achieve full 2pi phase coverage in reflection for far-field wavefront manipulation and evaluate the beam steering capability.
They also show the abilities of such structures to control SPP directional excitation in the near field by introducing birefringence in the design.
The article is well written and properly referenced, and the simulation methods appear sound and correctly applied.
The topic is timely and appropriate for Photonics journal, and I believe the paper is of high quality and worthy of publishing.
I only ask the authors to complete the information provided in the paper from the point of view of the robustness of the solution they conceived.
As the authors employ feature sizes close enough to the minimum available by standard fabrication methods, I believe it is important that they evaluate the robustness to inaccuracies of their design.
In particular, what are the most critical parameters in terms of their sensitivity to fabrication tolerances?

Author Response

The authors design and numerically simulate topological insulator-based gap-plasmon metasurfaces as a new material class for plasmonic applications as holograms, lenses, and other devices.
They use chalcogenide materials to achieve higher figures of merit in visible and UV bands than metals in a metal-insulator-metal nanobrick metasurface.
By employing the PB phase principle, they achieve full 2pi phase coverage in reflection for far-field wavefront manipulation and evaluate the beam steering capability.
They also show the abilities of such structures to control SPP directional excitation in the near field by introducing birefringence in the design.
The article is well written and properly referenced, and the simulation methods appear sound and correctly applied.
The topic is timely and appropriate for Photonics journal, and I believe the paper is of high quality and worthy of publishing.

Reply 1: We thank the reviewer for the positive confirmation of the quality and the contemporary relevance of our work and the recommendation of publishing our manuscript in Photonics.

I only ask the authors to complete the information provided in the paper from the point of view of the robustness of the solution they conceived.
As the authors employ feature sizes close enough to the minimum available by standard fabrication methods, I believe it is important that they evaluate the robustness to inaccuracies of their design.
In particular, what are the most critical parameters in terms of their sensitivity to fabrication tolerances?

Reply 2: We thank the reviewer for pointing out the importance of evaluating the robustness to inaccuracies of our design due to the relatively small feature sizes close to the minimum available sizes by standard fabrication methods. Fabrication inaccuracies mainly occur for the length and width of the nanobricks since these parameters are set by lithography which typically has a much lower accuracy than the thickness of TI films made from either chemical vapour deposition or atomic layer deposition. Therefore, we added the following paragraph and Fig. S3:

Due to the rather small feature sizes of our designed TI metasurface, fabrication im-perfections must be considered. Variations in the height of the different layers can be ne-glected since film deposition either from chemical vapor deposition or atomic layer depo-sition has an accuracy down to [17]. Length and width of the nanobricks are set by lithog-raphy, which might result in larger deviations. Fig. S3 shows the cross- and co-polarized reflectance for the structure used in Fig. 3a and for 10nm detuned lengths and widths. Due to its smaller size the detuning of the width has a higher impact on the efficiency than the length. Here a 10nm reduced width yields the lowest conversion efficiency, 3% smaller at 500nm than the optimized one. This indicates a robust metasurface design with a high tolerance against fabrication imperfections. Note that the 180nm long brick yields a higher cross-polarization efficiency than the 170nm long one. However, the latter was used in Fig. 3 with sufficiently large gaps between the bricks to ensure fabrication feasibility.

Figure S3. In (a) the length L and in (b) the width W of the nanobrick metasurface used in Fig. 3a was detuned by ±10nm to test the stability against fabrication imperfections. For the original and detuned structures, the cross- and co-polarized reflectance amplitude is indicated with solid and dashed lines, respectively. In (a) the efficiency increases with the nanobrick length, however, longer bricks lead to smaller gaps between them. To keep a balance between efficiency and fabrication feasibility, a length of 170nm was chosen for the final design. The cross-polarization efficiency at 500nm is decreased by 2% for a 10nm smaller length. Furthermore, it is the highest for the chosen width of 45nm and the lowest with 3% less for a 35nm width (b).

Overall, the nanobrick width is the most critical parameter in terms of its sensitivity to fabrication imperfections. Longer bricks yield even better results, however, the interparticle gap will get smaller and hence increase fabrication difficulty due to the proximity effect. An additional increasing the pitch is not suitable since it will also lower the efficiency as well as the maximal achievable anomalous reflectance angle of the metasurface, see Fig. 3e. The influence of length and width variations on phase and reflectance for the structure used in Fig. 4 are already shown in Fig. 4b and Fig. S3, respectively.

Round 2

Reviewer 2 Report

The final version of the manuscript is suitable for publication, after revisions they addressed all points.